# Six-month follow-up of a booster dose of CoronaVac in two single-centre phase 2 clinical trials

Qianqian Xin[1,8], Qianhui Wu[2,8], Xinhua Chen[2,8], Bihua Han[3,8], Kai Chu[4,8], Yan Song[5], Hui Jin[6], Panpan Chen[6], Wanying Lu[2], Tuantuan Yang[1], Minjie Li [3,9 ✉], Yuliang Zhao [3,9 ✉], Hongxing Pan [4,9 ✉], Hongjie Yu [2,9 ✉] & Lin Wang [7,9 ✉]

Determining the duration of immunity induced by booster doses of CoronaVac is crucial for informing recommendations for booster regimens and adjusting immunization strategies. In two single-centre, double-blind, randomised, placebo-controlled phase 2 clinical trials, immunogenicity and safety of four immunization regimens are assessed in adults aged 18 to 59 years and one immunization regimen in adults aged 60 years and older, respectively. Serious adverse events occurring within 6 months after booster doses are recorded as pre-specified secondary endpoints, geometric mean titres (GMTs) of neutralising antibodies one year after the 3-dose schedule immunization and 6 months after the booster doses are assessed as pre-specified exploratory endpoints, GMT fold-decreases in neutralization titres are assessed as post-hoc analyses. Neutralising antibody titres decline approximately 4-fold and 2.5-fold from day 28 to day 180 after third doses in adults aged 18–59 years of age and in adults aged 60 years and older, respectively. No safety concerns are identified during the follow-up period. There are increases in the magnitude and duration of humoral response with homologous booster doses of CoronaVac given 8 months after a primary two-dose immunization series, which could prolong protection and contribute to building our wall of population immunity. Trial number: NCT04352608 and NCT04383574.

[1] Sinovac Biotech, Beijing, China. [2] School of Public Health, Fudan University, Key Laboratory of Public Health Safety, Ministry of Education, Shanghai, China. [3] Hebei Provincial Center for Disease Control and Prevention, Shijiazhuang, Hebei, China. [4] Jiangsu Provincial Center for Disease Control and Prevention, Nanjing, China. [5] Suining County Center for Disease Control and Prevention, Jiangsu, China. [6] Renqiu Center for Disease Control and Prevention, Hebei, China. [7] Sinovac Life Sciences, Beijing, China. [8]These authors contributed equally: Qianqian Xin, Qianhui Wu, Xinhua Chen, Bihua Han, Kai Chu. [9]These authors jointly supervised this work: Minjie Li, Yuliang Zhao, Hongxing Pan, Hongjie Yu, Lin Wang. ✉email: liminjie507@163.com; yuliang_zh1@163.com; panhongxing@126.com; yhj@fudan.edu.cn; wanglin@sinovac.com

Due in part to waning immunity and diminished protection over time following primary immunisation[1–3], particularly against the Delta (B.1.617.2) variant of SARS-CoV-2, many countries and regions are experiencing surges in COVID-19 cases. Booster doses given at 6–8 months after a primary schedule have been shown to increase neutralisation antibody levels against wild-type virus and reduce the immunity gap between wild-type virus and variants of concern[4,5]. Extended primary immunisation series were recommended by the World Health Organisation[6], especially for those at high risk of severe COVID-19 disease, and booster-dose programmes have been initiated in dozens of countries. With gradual understanding of the epidemiological parameters and immune escape potential of the Omicron variant (B.1.1.529), it is of critical importance to assess the protection and persistence of protection that current COVID-19 vaccines can provide.

Interim study results suggest that rates of confirmed infection and severe illness caused by the Delta variant could be significantly reduced in the short term following booster doses[7,8]. However, no experimental data on the long-term kinetics of neutralisation titres have been reported, even though in-vitro neutralisation titres are important predictors of protection from SARS-CoV-2 variants[9,10]. As CoronaVac is a commonly used vaccine and is contributing to the fight against the pandemic, assessing the duration of immunity following booster-dose administration will be important for improving and updating immunisation strategies. The 3 μg dose is the licensed formulation, and an additional (third) dose is recommended to be offered 6 months after the two-dose primary schedule. We conducted a study to assess immune persistence after a homologous booster dose of CoronaVac given 8 months after the 2nd dose of a two-dose primary immunisation series in two population groups: adults aged 18–59 years and adults aged 60 years or older.

## Results

In phase-2 clinical trial among 600 healthy adults aged 18–59 years, 129 (92.8%) of 139 participants from cohort 1a-14d-2m and 126 (96.9%) of 130 participants from cohort 2a-28d-2m completed blood sampling to assess immune persistence for 1 year after dose 3 among those assigned a primary third dose. Separately, 135 participants in cohort 1b-14d-8m (95.7% of the 141 participants assigned a booster dose) and 124 participants in cohort 2b-28d-8m (95.4% of the 130 participants assigned a booster dose) completed blood sampling to assess immune persistence for 6 months after dose 3. In phase-2 clinical trial among a total of 350 healthy adults aged 60 years and older, 283 (93.4%) of 303 participants who received a booster dose from cohort 3-28d-8m completed a 6-month follow-up after dose 3. Supplementary Fig. 1 shows the trial profile. Baseline characteristics of participants are shown in the reports of main findings for these two trials[11–13]. Baseline demographic characteristics of participants who received third doses between the study groups were similar (Supplementary Table 1).

There were 141 minor protocol deviations in cohort 1b-14d-8m and 1 minor protocol deviation in cohort 3-28d-8m that did not result in the exclusion of participants from the analysis, including 141 participants in cohort 1b-14d-8m who were given third doses 9–11 days outside of the pre-specified time window, and 1 participant in cohort 3-28d-8m who was given a second dose 5 days outside of the pre-specified time window (Supplementary Fig. 1).

Compared to antibody concentrations on day 28 after the booster dose, neutralization titer declined 3–4-fold by 6 months after the booster dose, which was given 8 months after a two-dose primary vaccination regimen in adults aged 18–59 years. In the 3 μg group in cohort 2b-28d-8m, GMTs decreased from 143.3 (95% CI 112.3–182.8) on day 28 to 36.4 (95% CI 28.7–46.1) on day 180 after a booster dose (Fig. 1 and Supplementary Fig. 2, Supplementary

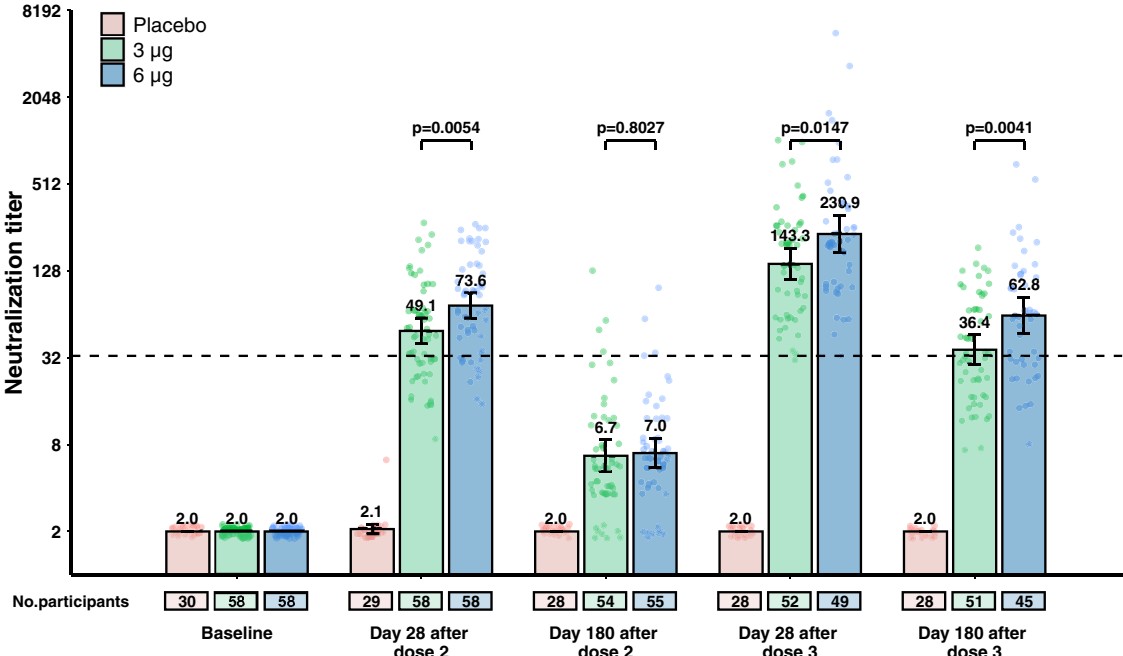

**Fig. 1 Neutralising antibody levels to ancestral SARS-CoV-2 in cohort 2b-28d-8m (adults aged 18–59 years old).** The number of participants for each group (placebo group, pink; 3 μg group, green; 6 μg group, blue) at each visit included in the analysis is provided below the bars. Dots are reciprocal neutralising antibody titres for individuals in the per-protocol population. Numbers above the bars are geometric mean titres (GMTs), and error bars indicate 95% CIs. GMTs and corresponding 95% CIs were calculated on the basis of standard normal distributions of log-transformed antibody titres. Numbers above the short horizontal lines are p values for comparisons between 3 μg group and 6 μg group using group *t* tests with log-transformation (two-sided). Titres lower than the limit of detection (1:4) are presented as half the limit of detection. The dotted horizontal line represents the protective threshold (1:33).

Table 2). With the exception of baseline and day 180 after dose 2, GMTs at other timepoints in cohort 2b-28d-8m were significantly higher in the 6 μg group than in the 3 μg group (Supplementary Fig. 2 and Supplementary Table 2). GMTs decreased from 137.9 (95% CI 99.9–190.4) on day 14 to 33.4 (95% CI 25.0–44.6) on day 180 after booster doses in cohort 1b-14d-8m (Fig. 1, Supplementary Fig. 2 and Supplementary Table 2). GMTs in cohort 1b-14d-8m on day 180 after the booster dose were significantly higher ($P = 0.02$) in the 6 μg group than in the 3 μg group; there were no significant differences between the two-dose amounts at other timepoints (Supplementary Fig. 2 and Supplementary Table 2). Regardless of the interval between the first two doses and antigen amount, by 1 year after a primary third dose, GMTs in vaccination groups were all at least twofold above the detection limit in cohort 1a-14d-2m and cohort 2a-28d-2m. There were no significant differences in GMTs between the 3 μg groups and the 6 μg groups at 1 year after dose 3 in the two cohorts (Supplementary Fig. 2 and Supplementary Table 2).

A similar pattern was observed in cohort 3-28d-8m, in which neutralisation titres declined from 158.5 [95% CI 96.9–259.1] on day 28 to 53.2 [95% CI 39.7–71.1] on day 180. GMTs on day 180 after the booster dose were highest in the 6 μg group (GMT 91.2 [95% CI 71.5–116.3], $P < 0.0001$), followed by the 3 μg group (Fig. 2 and Supplementary Table 3). In the 3 μg group and the 6 μg group, GMTs 6 months after booster doses among older adults (60 years and older) were numerically higher than among younger adults (18–59 years old), but without a statistical difference ($P = 0.05$). Results of sensitivity analyses showed that the use of average dilutions has no significant impact on the values of neutralisation antibody titre (Supplementary Tables 4–7).

GMT fold decreases during the 6 months after primary two doses, primary three doses, and booster doses were compared among vaccination groups, calculated as the ratio of GMT on day 28 to

GMT on day 180 after the specific dose. Taking the 3 μg group in cohort 2b-28d-8m as an example, the GMT fold decrease between day 28 and day 180 after a booster dose (4.1-fold) was significantly lower than that observed between day 28 and day 180 after the second dose (6.8-fold; $P = 0.0007$; Fig. 3), which was numerically lower than that of day 28 and day 180 after primary three doses in cohort 2a-28d-2m (4.9-fold; $P = 0.35$; Supplementary Fig. 3). GMT fold decreases between day 28 and day 180 after the second dose were similar in cohort 1b-14d-8m (7.3-fold) and cohort 2b-28d-8m (6.8-fold; $P = 0.75$; Supplementary Fig. 3), regardless of the interval of first two doses. Likewise, the GMT fold decrease between day 28 and day 180 after the booster dose (2.5-fold) was significantly lower than that between day 28 and day 180 after the second dose (10.7-fold; $P < 0.0001$; Fig. 3). Compared with adults aged 18–59 years old (cohort 2b-28d-8m), the GMT fold decrease was greater in adults aged 60 years and older (cohort 3-28d-8m) after primary two doses (6.8-fold vs 10.7-fold, $P = 0.03$), but was lower after booster doses (4.1-fold vs 2.5-fold, $P < 0.0001$; Fig. 3). There were no significant differences in GMT fold decreases between the 3 μg groups and 6 μg groups after booster doses among all the vaccination groups, irrespective of vaccination schedules and age grouping (Fig. 3, Supplementary Fig. 3 and Supplementary Fig. 4).

Serious adverse events that occurred from the beginning of immunisation to 6 months after second doses in cohort 1b-14d-8m, cohort 2b-28d-8m, and cohort 3-28d-8m, and that occurred from the beginning of immunisation to 6 months after third doses in cohort 1a-14d-2m and cohort 2a-28d-2m have been reported previously[13]. During the 6-month follow-up after booster doses in cohort 1b-14d-8m, cohort 2b-28d-8m, and cohort 3-28d-8m, serious adverse events were reported in one (2%) of 52 participants in the 3 μg group in cohort 2b-28d-8m (Supplementary Table 8), in four (5%) of 85 participants in the

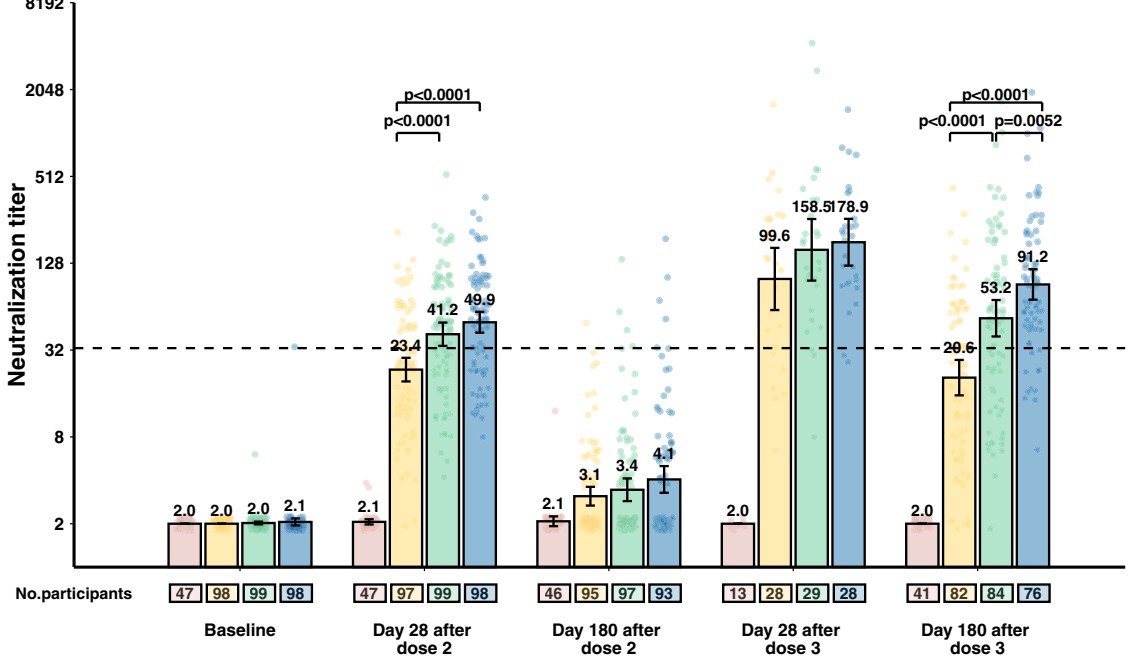

**Fig. 2 Neutralising antibody levels to ancestral SARS-CoV-2 in cohort 3-28d-8m (adults aged 60 years and older).** The number of participants for each group (placebo group, pink; 1.5 μg group, yellow; 3 μg group, green; 6 μg group, blue) at each visit included in the analysis is provided below the bars. Dots are reciprocal neutralising antibody titres for individuals in the per-protocol population. Numbers above the bars are geometric mean titres (GMTs), and error bars indicate 95% CIs. GMTs and corresponding 95% CIs were calculated on the basis of standard normal distributions of log-transformed antibody titres. Numbers above the short horizontal lines are $p$ values for comparisons between the 1.5 μg group, the 3 μg group and the 6 μg group using ANOVA models with log-transformation. Bonferroni correction done as a post hoc test if the variance was significant. Only $P$ values indicating significant differences are marked. Titres lower than the limit of detection (1:4) are presented as half the limit of detection. The dotted horizontal line represents the protective threshold (1:33).

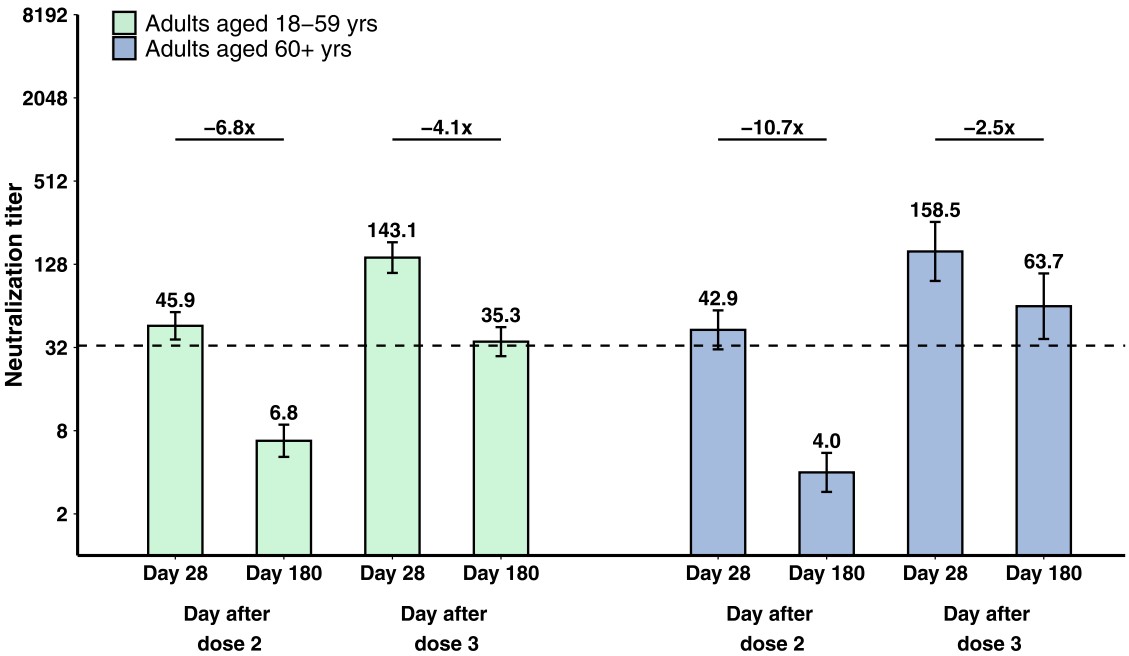

**Fig. 3 Decline in neutralising antibodies to ancestral SARS-CoV-2 in 3 μg groups in cohort 2b-28d-8m and cohort 3-28d-8m.** The number of participants with paired samples for adults aged 18–59 (green) and adults aged 60 and over (blue) was 49 and 29, respectively. Numbers above the bars are geometric mean titres (GMTs), and error bars indicate the 95% CIs. The dotted horizontal line represents the protective threshold (1:33). Numbers above the short horizontal lines are pairwise fold-change values. GMTs and corresponding 95% CIs were calculated on the basis of standard normal distributions of log-transformed antibody titres. GMT fold decreases in neutralisation titre were calculated as ratios of paired sera at two visits. Comparisons between groups were conducted by group $t$ tests with log-transformation (two-sided). $P$ values of pairwise comparisons were $P < 0.0001$, $P < 0.0001$, $P < 0.0001$, $P = 0.0187$, from left to right, respectively.

1.5 μg group, in five (6%) of 90 in the 3 μg group, in three (4%) of 81 in the 6 μg group, and in two (4%) of 47 in the placebo group in cohort 3-28d-8m (Supplementary Table 9). No participant in cohort 1b-14d-8m reported a serious adverse event. No serious adverse event in either trial was considered by the investigators to be related to vaccination, and no pre-specified trial-halting rules were met.

## Discussion

Following a primary three-dose regimen for immunisation, neutralisation antibody levels declined 6 months later and remained stable during the next 6 months. Neutralisation antibody levels were substantially increased by booster doses given 8 months after primary two-dose regimens and were maintained over the following 6 months—comparable with levels after primary two-dose immunisation regimens. When booster doses were given 8-month after primary immunisation, the decay rates of neutralisation titres over the 6 months after booster-dose administration were much slower than that after primary two-dose regimens regardless of age group and antigen amount. Memory B cells are known to proliferate and produce antibodies that maintain immunity after repeated exposure to antigens—a phenomenon that likely maintains protection and contributes to building our wall of population immunity.

Observed GMT fold decreases during the 6 months following primary two doses in the two age groups in our CoronaVac study were in line with results of a BNT162b2[4] vaccine study, which were 6.0-fold in 18–55 years of age and 13.1-fold in 65–85 years of age from 7 days after dose 2 to before dose 3 (7.9–8.8 months after dose 2). We found that GMT fold decreases after booster doses were lower and neutralisation titres 6 months after booster doses were numerically higher in adults aged 60 years and older (cohort 3-28d-8m) than in adults aged 18–59 years old (cohort 2b-28d-8m), which is in

contrast with common sense that immune responses to vaccination are generally weaker in older adults[14]. Notably, differences were small and there was overlap between younger adults and older adults in neutralising activity against SARS-CoV-2 viruses in the mRNA-1273 vaccine recipients[15]. Age did not appear to compromise antibody response, even after accounting for severity among COVID-19 patients[16]. More experimental data are required to address the age heterogeneity of long-term neutralisation dynamics following vaccination.

A meta-analysis that summarised immune escape potential of different SARS-CoV-2 variants against immunity induced by both natural infection and vaccination showed that the average fold reduction of neutralising antibody level against the Delta variant was 2.4 (95% CI: 1.1–5.2) for inactivated vaccines in live virus neutralisation assays when compared to that of prototype strains[17]. However, for individuals vaccinated with CoronaVac, the average reduction against Delta was 9.2-fold compared with the prototype strain using authentic virus neutralisation assay[18]. Currently, there are no available data for immune evasion of the humoral immunity elicited by inactivated vaccines for the recently emerged Omicron variant. One report showed that sera from individuals who received two doses of BNT162b2 exhibited an average 25-fold reduction in neutralisation titres against the Omicron variant compared to wild-type virus when using a pseudovirus neutralisation test[19]. Another study showed a higher fold reduction of 41.4 for the Omicron variant among individuals with previous infection or vaccination[20].

Even though neutralisation titres induced by COVID-19 vaccines decline over time and against variants, vaccine effectiveness against severe COVID-19 illness is sustained, including against severe outcomes caused by Delta[21]. Although the limited available evidence shows that the immune escape of Omicron is significant, vaccine effectiveness against hospitalisation may be well maintained. Booster vaccination with current vaccines increases the

affinity of antibody and neutralisation potency better than that achieved with primary vaccination only, and this effect can likely be predicted to provide robust protection from severe infection outcomes from the current SARS-CoV-2 variants of concern[9]. One recent study reported that a moderate to high vaccine effectiveness against mild infection of 70–75% was seen in the early period after a booster dose of BNT162b2 following either ChAdOx1-S or BNT162b2 as a primary series, despite the longer intervals after primary vaccination[22], underscoring the necessity of timely administration of a booster dose.

Higher antigen content appeared to induce higher neutralisation titres and maintain higher levels in the 6 months of follow-up after booster doses in the medium term. This implies that vaccines containing higher antigen content (i.e. 6 µg) could be considered for booster immunisation programmes. Heterologous booster vaccination has been shown to induce strong humoral responses and augment neutralisation potency[23,24]. At 4–8 months after primary immunisation with CoronaVac, a significantly higher degree of humoral immunogenicity against the prototype strain and the Gamma, Beta, and Delta variants was observed following a third dose of ZF2001 (a protein subunit vaccine manufactured by Anhui Zhifei Longcom Biopharmaceutical)[18]. A heterologous prime-boost regimen with Convidecia (a type-5-adenovirus-vectored COVID-19 vaccine manufactured by CanSino) after priming with CoronaVac 3–6 months earlier induced approximately 5.9-fold higher live virus neutralising antibodies than homologous boosting induced[25]. By identifying various forms of antigens from vaccines made on different platforms, the immune system apparently can be trained to produce a more balanced and comprehensive immune response that enhances the effect of current vaccines through heterologous immunisation strategies. Heterologous boosting has clear policy implications, as it can provide solutions to curb the pandemic of emerging variants before developing new vaccines. It should be noted there are no large-scale heterologous immunisation practices until recently, and more high-quality safety and effectiveness research evidence is required to improve immunisation strategies. In addition, several research studies have shown that extended dosing intervals generate more favourable immune responses[5,26]. "Mix-and-match" regimens and longer dosing interval strategies may also be helpful in lower-income countries, where some vaccines may be in short supply some of the time. With much of the world yet to be vaccinated, re-doubling our efforts for equitable and speedy vaccine delivery on a global scale and improving initial vaccination coverage should be our primary focus.

Immune memory is what leads to long-term immunity, but it is difficult to predict how long immunity will last because the exact mechanisms of protective immunity against SARS-CoV-2 or COVID-19 are still not clear. The 6-month marker in our study is an important milestone, but long-term immune response and effectiveness need to be continuously monitored into the foreseeable future.

Our study has several limitations. First, T-cell responses and neutralisation tests in vitro against emerging variants were not assessed in our study; these should be further explored. Second, multicentre studies will be needed to assess primary outcomes among subpopulations for whom our study had relatively small proportions, for example, people with multiple underlying conditions or immunosuppressive conditions. Third, the follow-up time of our study is relatively short. However, timely reporting of follow-up data is very important for ongoing adjustment of immunisation strategies in the context of a pandemic with frequent emergence of variants. Fourth, although neutralising antibodies are related to protection, actual protection from infection with current and future variants will need to be monitored with real-world observational studies. Further research to identify correlates of protection is essential.

In conclusion, a homologous booster dose of CoronaVac given 8 months after 2nd dose of the primary two-dose immunisation recalls robust neutralisation antibody levels and significantly delays antibody attenuation in adults aged 18 years and older. More experimental and long-term monitoring data are needed to optimise the selection of booster doses and booster-dose intervals to most effectively combat the pandemic.

## Methods

**Study design and participants.** The study designs and methods for these two phase II trials have been previously reported[11]. Key exclusion criteria for trial enrolment included suspected or laboratory-confirmed SARS-CoV-2 infections and known allergy to any vaccine component. A complete list of exclusion criteria is in the protocol in Supplementary Material. All participants gave written informed consent to participate in the study before administration of first doses and booster doses. The two trials were registered with ClinicalTrials.gov, NCT04352608 and NCT04383574.

Briefly, the initial trial involving 600 healthy adults aged 18–59 years old in a single-centre, double-blind, randomised, placebo-controlled, phase-2 clinical trial conducted from May 3, 2020 in Suining county, Jiangsu province, China. Following enrollment, participants were randomised to receive three doses of either 3 µg of CoronaVac, 6 µg of CoronaVac, or placebo with an interval of 14 days or 28 days between the first two doses and 2 months or 8 months between the second and third doses; the respective study groups were cohort 1a-14d-2m, cohort 1b-14d-8m, cohort 2a-28d-2m, and cohort 2b-28d-8m. One hundred fifty participants were assigned to each cohort, and 3 µg or 6 µg of CoronaVac or placebo were randomly assigned in a 2:1:1 allocation ratio.

The other trial, involving 350 healthy adults aged 60 years and older, was a single-centre, double-blind, randomised, placebo-controlled, phase-2 clinical trial conducted from June 12, 2020 in Renqiu county, Hebei province, China. Following enrollment, participants were randomised to receive three doses of 1.5, 3 or 6 µg of CoronaVac or placebo with an interval of 28 days between the first two doses and 8 months between second and third doses; this study group is cohort 3-28d-8m. Randomisation was performed with a 2:2:2:1 allocation ratio.

Electronic Data Capture (EDC) RIEHEN (Version: 2.1.1608) was used to establish the electronic Case Report Form (eCRF) in both trials to record clinical trial data. Information was inputted with standard language according to the EDC instructions and eCRF filling instructions. Randomisation codes for each vaccination schedule cohort were generated individually and randomly assigned using block randomisation developed with SAS version 9.4. Adults aged 18–59 years were assigned with a block size of five and adults aged 60 years and older were assigned with a block size of fourteen. Concealed random group allocations and blinding codes were kept in signed and sealed envelopes. Investigators, participants, and laboratory staff were masked to group assignment. The randomisation code was assigned to each participant in sequence in the order of enrolment by investigators, who were involved in the rest of the trial.

**Follow-up.** Essential steps and timing for each visit specified in the protocol are shown in Supplementary Visit Plan. Conditions leading to participant withdrawal and suspension criteria were reported previously[11], including unacceptable adverse events, abnormal clinical manifestations, participants' request. Participants who received primary third doses 2 months after the second dose (cohort 1a-14d-2m and cohort 2a-28d-2m) had blood samples drawn 1 year after the third dose to evaluate immune persistence of this three-dose primary immunisation regimen. Participants who received booster doses 8 months after the second dose (cohort 1b-14d-8m, cohort 2b-28d-8m and cohort 3-28d-8m), had blood samples drawn 6 months after the booster dose to evaluate immune persistence of this booster regimen.

Immunological assessment methods and related procedures are described in the Supplementary Neutralisation Assay. Neutralising antibodies against infectious SARS-CoV-2 (virus strain SARS-CoV-2/human/CHN/CN1/2020, GenBank accession number MT407649.1, https://www.ncbi.nlm.nih.gov/nuccore/MT407649.1) were quantified using a microcytopathogenic effect assay. We treated the neutralising antibody titer of the serum specimen as the reciprocal of the average dilutions of two wells when one of two adjacent wells was pathological while the other not. To avoid the use of the average would not deflate or inflate the values of neutralisation antibody titre, we conducted sensitivity analyses only to adopt higher dilutions or lower dilutions respectively. Serious adverse events were recorded for 6 months after the third dose for participants in every cohort. Serious adverse events were coded by the Medical Dictionary for Regulatory Activities (MedDRA) System Organ Class. The existence of causal associations between adverse events and vaccination was determined by the investigators.

**Outcomes.** A complete list of study endpoints is provided in the Supplementary Study Endpoints. Results as of 28 days after booster doses (for cohort 1b-14d-8m, cohort 2b-28d-8m and cohort 3-28d-8m) and 6 months after primary three doses (for cohort 1a-14d-2m and cohort 2a-28d-2m) have been reported previously[11]. Here, we report the follow-up immunogenic results including geometric mean titres (GMTs) of neutralising antibodies to infectious SARS-CoV-2 one year after

the full schedule immunisation (for cohort 1a-14d-2m and cohort 2a-28d-2m), 6 months after the booster dose (for cohort 1b-14d-8m, cohort 2b-28d-8m, and cohort 3-28d-8m), all of which are pre-specified exploratory endpoints. As did Khoury and colleagues[10], we used a protective threshold of 33 for CoronaVac vaccine, which was defined as the neutralisation titer at which an individual will have a 50% protective efficacy for CoronaVac. Titres lower than the limit of detection (1:4) were treated as half the limit of detection.

Serious adverse events occurring within 6 months after booster doses (for cohort 1b-14d-8m, cohort 2b-28d-8m and cohort 3-28d-8m) were recorded; serious adverse events were pre-specified secondary endpoints. Comparisons of GMT fold decreases in neutralisation titres within 1 year after full-course vaccination for cohort 1a-14d-2m and cohort 2a-28d-2m, and within 6 months after second doses and third doses for cohort 1b-14d-8m, cohort 2b-28d-8m, and cohort 3-28d-8m, were post hoc analyses. Given that the 3 μg dose is the licensed formulation and an additional (third) dose is recommended to be offered 6 months after the two-dose primary schedule, we present results for the 3 μg groups in cohort 2b-28d-8m and cohort 3-28d-8m in the main text and provide detailed results for other intervention groups in the Supplementary.

**Ethical statement**. We complied with all relevant ethical rules. The complete study protocol for adults aged 18–59 years old was approved by the ethics committees of Jiangsu Provincial Centre for Disease Control and Prevention (JSJK2020-A021-02), and the complete study protocol for adults aged 60 years and older was approved by the ethics committees of Hebei Provincial Centre for Disease Control and Prevention (IRB2020-006).

**Statistical analysis**. The sample size was determined following requirements of the National Medical Products Administration, China's regulatory authority for vaccines. We assessed immunological endpoints in the per-protocol population, which included all participants who completed their assigned doses and had antibody results available according to the protocol. Serious adverse events were evaluated in the safety population for booster-dose groups, which included all participants who received a booster dose of the study vaccine. GMT fold decreases in neutralisation titres were assessed among participants who received three doses and had antibody results from all visits.

Pearson $\chi^2$ test or Fisher's exact test were used to analyse categorical outcomes. We calculated 95% CIs for categorical outcomes using the Clopper–Pearson method.

We calculated GMTs and corresponding 95% CIs on the basis of standard normal distributions of log-transformed antibody titres. GMT fold decreases in neutralisation titres were calculated as ratios of paired sera at two visits. ANOVA models with log-transformation were used to detect differences among groups. Comparisons were done between groups by group $t$ tests with log-transformation and Bonferroni correction done as a post hoc test if the variance was significant. Hypothesis testing was two-sided, and we considered $P$ values of less than 0.05 to be significant. We used R software version 4.0.2 for all analyses.

**Reporting summary**. Further information on research design is available in the Nature Research Reporting Summary linked to this article.

## Data availability

The study protocols are available in the Supplementary Material. To protect participants' confidentiality, the individual participant data that underlie the results reported in this article (text, tables, figures and extended data) will only be shared after de-identification. Due to the clinical trial in adults aged 60 years and older is ongoing, in order to maintain the blind status of this trial, the data will be available following clinical study report (CSR) of immune persistence analysis (September 2022). Researchers who provide a scientifically sound proposal will be allowed access to the individual participant data. Proposals should be directed to wanglin@sinovac.com.

## Code availability

The R code for the main analysis is available on GitHub at https://github.com/cxhhhh24/sinoVac_antibody.

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

## Acknowledgements

We thank Dr. Lance Rodewald from the Chinese Center for Disease Control and Prevention for his comments and English language editing. The study was supported by grants from

National Key Research and Development Program (2020YFC0849600), Beijing Science and Technology Program (Z201100005420023), and Key Program of the National Natural Science Foundation of China (82130093). The funders had no role in study design, data collection and analysis, decision to publish or preparation of the manuscript.

## Author contributions

Q.X., Q.W., X.C., B.H., K.C., M.L., Y.Z., H.P., H.Y. and L.W. formulated the study design, and performed the data collection, analysis, interpretation and writing of the manuscript. H.J., P.C. and Y.S. collected the data and revised the manuscript. T.Y. carried out the laboratory assays and revised the manuscript. W.L. analysed the data and revised the manuscript. All authors had full access to all of the data (including statistical reports and tables) in the study and can take responsibility for the integrity of the data and the accuracy of the data analysis.

## Competing interests

H.Y. has received research funding from Sanofi Pasteur, GlaxoSmithKline, Yichang HEC Changjiang Pharmaceutical Company, and Shanghai Roche Pharmaceutical Company. None of those research funding is related to the development of COVID-19 vaccines. Q.X. and T.Y. were the employees of Sinovac Biotech Ltd., L.W. was an employee of Sinovac Life Sciences Co., Ltd. The remaining authors declare no competing interests.
