## [Peer Review File · Nature Communications]

Six-month follow-up of a booster dose of CoronaVac in two single-centre phase 2 clinical trialsReviewers' Comments:

Reviewer #1:

Remarks to the Author:

The authors assessed and reported neutralizing antibody level for longitudinal blood samples following a 3-dose primary series or booster after a 2-dose primary series. The data showed that the booster given at 8 months after the 2-dose primary series increased neutralizing antibody level, and had a slower decay when compared to data from post primary series. This report is important as booster has been shown to be necessary to maintain high VE against hospitalization with Omicron and CoronaVac has been widely administered across the globe.

As a center piece for this manuscript, the neutralization assay is poorly described in the supplemental material and a number of questions are listed below.

1. What's the assay status: preclinical, qualified or validated?
2. What cells and virus are used for the assay?
3. How the virus was propagated and quality controlled?
4. How is CCID50 determined, given 100 CCID50 is used for each well in the assay
5. On page 3 of the supplemental material, it says "For example, 2 wells with high dilution of 1:8 have a complete CPE, while the adjacent 2 wells with low dilution of 1:16 have no CPE". How could it be possible that the wells with low quantity of serum (1:16; more diluted) has no CPE, while the wells with high quantity of serum (1:8; less diluted) has CPE?
6. On the same page, it says "Micro cytopathic effect assay was adopted". Where is it adopted from and where is the reference?
7. In the assay description, it says the least diluted sample in the assay (after mixing with virus) is 1:8, i.e. 1:8 is the assay limit of detection. However:
 - a. There are many data points in all figures between values of 2 and 8. How were these data points generated?
 - b. In figure legend of Fig 1 and 2, it says "Titres lower than the limit of detection (1:4) are presented as half the limit of detection". The LOD definition in the figure is inconsistent from the method description.
 - c. Furthermore, in figure legend of Fig 1 and 2, it mentioned "seropositive threshold (1:8)". Seropositivity shall be defined by diagnostic assays, not be the LOD of this neutralization assay with unknown assay status.
 - d. What's the purpose of putting "Dotted horizontal lines represent the seropositive threshold (1:8) and four-fold seropositive threshold (1:32)" on the figure?

In the abstract, it says "improvement in the kinetics of the humoral response" while there is no evidence in the manuscript to support this conclusion.

Minor points:

1. "8 month after two-dose primary series" is misleading as the authors are counting from day 0 when the first dose was administered. It should be defined as the time from the 2nd dose.
2. In the manuscript, 3 difference doses of booster were assessed: 1.5, 3 and 6 ug. However, there is no mention about what's the authorized dose for this vaccine in the primary series.
3. Figure S1, Month# should be added into the diagram since the day# doesn't directly translate to the month# mentioned in the text.

Reviewer #2:

Remarks to the Author:

The authors describe the amplitude and durability of neutralising antibody titers obtained after a third

dose of CoronaVac either 2 months or 8 months after the second dose. Due to the ongoing SARS-CoV-2 pandemic which may be transitioning to endemic status, understanding boostability and durability of responses with CoronaVac is of interest to the SARS-CoV-2 vaccine field.

As commented on by the authors in the discussion, it is somewhat surprising that adults aged 60+ had higher titers after boosting than younger adults; likewise the decline is less than might be expected.

The primary limitation of the study is that while titers to the ancestral lineage of SARS-CoV-2 are presented, titers against variants of concern are not. As there is mounting evidence that titers are lower against VOCs with ancestral strain-based vaccines, this is an important element for understanding the data in context.

Additional minor comments:

- 1) Line 64 statement needs to be supported with a reference or softened to "a commonly used vaccine."
- 2) Line 96 clarity - replace dash with 3 to 4
- 3) Line 94-95 correct carriage return
- 4) Line 110 replace "durable" with "still elevated," likewise on line 167 replace "durable" with "maintained"
- 5) Line 254 - typo "continuously" not "continuous"
- 6) Lines 254-257 are political opinion. Please remove.

REVIEWER COMMENTS

Reviewer #1:

The authors assessed and reported neutralizing antibody level for longitudinal blood samples following a 3-dose primary series or booster after a 2-dose primary series. The data showed that the booster given at 8 months after the 2-dose primary series increased neutralizing antibody level, and had a slower decay when compared to data from post primary series. This report is important as booster has been shown to be necessary to maintain high VE against hospitalization with Omicron and CoronaVac has been widely administered across the globe.

We thank the reviewer for the positive assessment of our manuscript and for her/his many useful comments that helped improve the manuscript.

As a center piece for this manuscript, the neutralization assay is poorly described in the supplemental material and a number of questions are listed below.

1. What's the assay status: preclinical, qualified or validated?

We apologize for the lack of clarity for the assay detail. The microneutralization assay used in the trial had been validated. Specifically, for the internal validation, we evaluated the specificity of the microneutralization assay by using samples collected from animals or humans infected with other viruses, such as influenza virus, poliovirus, enterovirus, as well as serum obtained from COVID-19 convalescent patients. In addition, the assay was tested for applicability by using SARS-CoV-2-infected samples from varied species, and accuracy by repeated measurements with the results shown in the following tables. This neutralization assay has been validated to have good specificity, accuracy, and applicability.

Table 1. Validation of specificity of neutralization assay

Samples	Result	Positive/ Negative	Whether acceptable
Hepatitis A mab ascites	<1: 4	N	Yes

EV71 positive serum control	<1: 4	N	Yes
Poliovirus type I bovine serum	<1: 4	N	Yes
Influenza A(H1N1) positive serum control	<1: 4	N	Yes
Influenza A(H3N2) positive serum control	<1: 4	N	Yes
Influenza B positive serum control	<1: 4	N	Yes
Influenza A(H5N1) positive serum control	<1: 4	N	Yes
SARS-CoV-2 goat antiserum	1: 1024	P	Yes
COVID-19 convalescent patient	1: 96	P	Yes

Table 2 Validation of accuracy of neutralization assay

Staff	Sample	Neutralization titer (1:)			Geometric mean value (1:)	Reference range (1:)
		Day 1	Day 2	Day 3		
A	SARS-CoV-2 goat	768/768	1024/512	1536/512	768	384~1536
B	antiserum	768/512	768/384	1536/768	768	384~1536
Geometric mean value (1:)		768	512	768		/
Reference range (1:)		384~1536	256~1024	384~1536		

Table 3 Validation of applicability of neutralization assay

Sample	Result	Acceptable standard	Whether acceptable
Negative serum control	<1: 4	Negative	Yes
COVID-19 convalescent patient 1	96	Positive	Yes
COVID-19 convalescent patient 2	24	Positive	Yes
COVID-19 convalescent patient 3	24	Positive	Yes
Rabbit antiserum	256	Positive	Yes
Mouse antiserum	192	Positive	Yes
Mouse antiserum	48	Positive	Yes
Goat antiserum	384	Positive	Yes

Recombinant protein immunized rabbit antiserum	6144	Positive	Yes
---	------	----------	-----

2. What cells and virus are used for the assay?

We apologize for the lack of clarity and thank you for pointing this out to us. Vero cells from kidney tissues of adult African green monkey (W.H.O.VERO SEED LOT 10-87) and SARS-CoV-2/human/CHN/CN1/2020 (GenBank number MT407649.1) were used in the neutralization assay. We have checked the description of virus in the main text and added details of the Vero cells in the Supplementary (Supplementary Method, p2, lines 25-27).

The wording in the Methods section now states, “Neutralising antibodies against infectious SARS-CoV-2 (virus strain SARS-CoV-2/human/CHN/CN1/2020, GenBank accession number MT407649.1) were quantified using a microcytopathogenic effect assay.”

3. How the virus was propagated and quality controlled?

We apologize for the lack of clarity. SARS-CoV-2 virus was seeded and propagated in a monolayer of Vero cells by using Medium 199 with 2% FBS and 1% penicillin-streptomycin.

Cells were seeded in T175 flasks and the cell monolayer was washed with sterile phosphate buffered saline (PBS). After removal of the PBS, the cells were infected with 0.04mL/cm² of medium containing the virus. The cell-virus mixture was incubated at 37°C in a humidified atmosphere with 5% CO₂ for 5-10 days. The flasks were observed every two days and the virus was harvested when 70%- 90% of the cells manifested CPE. The flask was placed horizontally at -20±2°C for more than two hours to harvest the virus. After unfreezing propagated virus, the culture medium was centrifuged to remove cell debris, then aliquoted and stored at -70°C.

The details above have been added in Supplementary (Supplementary Method, p2, lines 28-36).

To ensure the virus load used in neutralization is appropriate, we conducted back titrations for each batch of neutralization assays. Based on standardized virus neutralization methods to measure antibody levels against poliovirus in human sera released by WHO [Polio laboratory manual 4th edition, 2004. World Health Organization, Geneva, Switzerland], we conducted back titrations to confirm the amount of virus used in each test was within the range of 32 - 320 CCID₅₀/50 µl.

4. How is CCID₅₀ determined, given 100 CCID₅₀ is used for each well in the assay

The virus was titrated in serial 10-fold dilutions (from 10⁻¹ to 10⁻⁸) to obtain a 50% cell culture infective dose (CCID₅₀) on 96- well culture plates of VERO cells. The 96-well plate was incubated at 36.5±1°C in a humidified atmosphere with 5% CO₂ for 5 days, and observed for the presence of CPE by means of an inverted optical microscope on day 5. The end- point titres were calculated according to the Behrens-Karber [Polio laboratory manual 4th edition, 2004. World Health Organization, Geneva, Switzerland] based on eight replicates for titration.

5. On page 3 of the supplemental material, it says “For example, 2 wells with high dilution of 1:8 have a complete CPE, while the adjacent 2 wells with low dilution of 1:16 have no CPE”. How could it be possible that the wells with low quantity of serum (1:16; more diluted) has no CPE, while the wells with high quantity of serum (1:8; less diluted) has CPE?

We apologize for this mistake and thank you for pointing this out to us. We have revised these sentences as follow (Supplementary Method, p2, lines 66-70):

“For example, 2 wells with high dilution of 1:16 have a complete CPE, while the adjacent 2 wells with low dilution of 1:8 have no CPE; or in 2 adjacent wells with dilutions of 1:8 and 1:16, one has a CPE, while the

other does not. In this case, the reciprocal of the average dilutions of 2 wells is the neutralizing antibody titer of the serum.”

6. On the same page, it says “Micro cytopathic effect assay was adopted”. Where is it adopted from and where is the reference?

This assay has been validated and the results have been peer reviewed. References are:

1. Wu Z, Hu Y, Xu M, et al. Safety, tolerability, and immunogenicity of an inactivated SARS-CoV-2 vaccine (CoronaVac) in healthy adults aged 60 years and older: a randomised, double-blind, placebo-controlled, phase 1/2 clinical trial. *Lancet Infect Dis.* 2021;21(6):803-812. doi:10.1016/S1473-3099(20)30987-7
2. Han B, Song Y, Li C, et al. Safety, tolerability, and immunogenicity of an inactivated SARS-CoV-2 vaccine (CoronaVac) in healthy children and adolescents: a double-blind, randomised, controlled, phase 1/2 clinical trial. *Lancet Infect Dis.* 2021;21(12):1645-1653. doi:10.1016/S1473-3099(21)00319-4
3. Zhang Y, Zeng G, Pan H, et al. Safety, tolerability, and immunogenicity of an inactivated SARS-CoV-2 vaccine in healthy adults aged 18-59 years: a randomised, double-blind, placebo-controlled, phase 1/2 clinical trial. *Lancet Infect Dis.* 2021;21(2):181-192. doi:10.1016/S1473-3099(20)30843-4
4. Zeng G, Wu Q, Pan H, et al. Immunogenicity and safety of a third dose of CoronaVac, and immune persistence of a two-dose schedule, in healthy adults: interim results from two single-centre, double-blind, randomised, placebo-controlled phase 2 clinical trials [published online ahead of print, 2021 Dec 7]. *Lancet Infect Dis.* 2021;S1473-3099(21)00681-2. doi:10.1016/S1473-3099(21)00681-2

7. In the assay description, it says the least diluted sample in the assay (after mixing with virus) is 1:8, i.e. 1:8 is the assay limit of detection. However:

- a. There are many data points in all figures between values of 2 and 8. How were these data points generated?

- b. In figure legend of Fig 1 and 2, it says “Titres lower than the limit of detection (1:4) are presented as half the limit of detection”. The LOD definition in the figure is inconsistent from the method description.
- c. Furthermore, in figure legend of Fig 1 and 2, it mentioned “seropositive threshold (1:8)”. Seropositivity shall be defined by diagnostic assays, not be the LOD of this neutralization assay with unknow assay status.
- d. What’s the purpose of putting “Dotted horizontal lines represent the seropositive threshold (1:8) and four-fold seropositive threshold (1:32)” on the figure?

We apologize for the lack of clarity and thank you for your comments. Specifically, serum was serially diluted from 1:4 to 1:8192 with four-fold dilutions. We define the neutralization titer as the reciprocal of the highest dilution without cytopathic effects for serially-diluted serum before adding equal volume of infectious SARS-CoV-2 virus. Hence, the lower limit of detection is 1:4 (the lowest dilution) and titers lower than the lowest dilution were assigned a titer of 2. Since World Health Organization (WHO) has not initiated standardized operation procedure of neutralization assay for SARS-CoV-2, some studies defined the lower limit of detection by using initial dilution of the original serum sample¹, while other studies used the final dilution of the serum-virus mixture². According to WHO-initiated SOP for other infectious disease (influenza virus and enterovirus) and a MERS neutralization assay protocol⁴, we choose to use the initial dilution of the original serum sample (before mixing with virus).

Therefore, for question a, as mentioned above, samples with neutralization titers lower than 1:4 were assigned values of 2. Samples with complete CPE in higher dilutions with no CPE in lower dilutions were assigned to an intermediate value between these two dilutions, which is why there were some titre values between 4 and 8.

For question b, the lower limit of detection for this assay is 1:4. We have revised this paragraph to make it clear that we use the initial dilution of serum sample rather than serum-virus mixture to serve as the lower limit of detection as follow (Supplementary Method, p2-3, lines 51-52):

“The initial dilution of serum sample rather than serum-virus mixture was used to serve as the lower limit of detection (1:4).”

For question c, there are two conceptually different approaches for setting the seropositivity threshold. One approach requires obtaining reference samples of seronegatives and seropositives, after which a threshold that appropriately balances sensitivity and specificity is chosen and used for future samples. A second approach requires no reference samples and instead uses a 2-component Gaussian Mixture Model (GMM) to estimate the underlying seronegative and seropositive distributions, from which the threshold is chosen. Since the standard operating procedures for neutralization test against SARS-CoV-2 has not been established, we defined the four-fold of lowest assigned neutralization titer (2) as the positive threshold according to experience with influenza vaccine. With the deepening of research on correlates of protection, a logit correlation between neutralization titers and protection level was established by *Khoury et al (Nat Med, 2021)*, which denoted that the neutralizing antibody titer of 33 induced by CoronaVac could provide 50% protection against symptomatic infections.

For question d, the two dotted horizontal lines represent the seropositivity threshold (1:8) and protective threshold respectively (1:33), which were estimated by Khoury et al⁵. We have added clear definitions in the Method section as follows, please see lines 361-364:

“As did Khoury and colleagues, we used a protective threshold of 33 for CoronaVac vaccine, which was defined as the neutralization titer at which an individual will have a 50% protective efficacy for

CoronaVac.”

References

1. Xia S, Zhang Y, Wang Y, et al. Safety and immunogenicity of an inactivated COVID-19 vaccine, BBIBP-CorV, in people younger than 18 years: a randomised, double-blind, controlled, phase 1/2 trial. *Lancet Infect Dis.* 2022;22(2):196-208. doi:10.1016/S1473-3099(21)00462-X
2. Anderson EJ, Roupael NG, Widge AT, et al. Safety and Immunogenicity of SARS-CoV-2 mRNA-1273 Vaccine in Older Adults. *N Engl J Med.* 2020;383(25):2427-2438. doi:10.1056/NEJMoa2028436
3. Global Influenza Programme WEP, Global Influenza Surveillance and Response System. Manual for the laboratory diagnosis and virological surveillance of influenza. ISBN: 9789241548090
4. Koch T, Dahlke C, Fathi A, et al. Safety and immunogenicity of a modified vaccinia virus Ankara vector vaccine candidate for Middle East respiratory syndrome: an open-label, phase 1 trial. *Lancet Infect Dis.* 2020;20(7):827-838. doi:10.1016/S1473-3099(20)30248-6
5. Khoury DS, Cromer D, Reynaldi A, et al. Neutralizing antibody levels are highly predictive of immune protection from symptomatic SARS-CoV-2 infection. *Nat Med.* 2021;27(7):1205-1211. doi:10.1038/s41591-021-01377-8

8. In the abstract, it says “improvement in the kinetics of the humoral response” while there is no evidence in the manuscript to support this conclusion.

Thanks very much for your comment. We have removed this statement from the abstract.

Minor points:

1. “8 month after two-dose primary series” is misleading as the authors are counting from day 0 when the first dose was administered. It should be defined as the time from the 2nd dose.

Thanks very much for pointing out a mistake in the text. We revised the descriptions as “8 month after the 2nd dose” through the paper.

2. In the manuscript, 3 difference doses of booster were assessed: 1.5, 3 and 6 ug. However, there is no mention about what’s the authorized dose for this vaccine in the primary series.

Thanks very much for your comment. We have mentioned that the 3 ug was the licensed formulation in Method section (Line 373). We added this information in the Introduction Section to address this issue, please see lines 71-73:

“The 3 μg dose is the licensed formulation, and an additional (third) dose is recommended to be offered 6 months after the two-dose primary schedule.”

3. Figure S1, Month# should be added into the diagram since the day# doesn’t directly translate to the month# mentioned in the text.

Thanks very much for your comment. We checked the diagram and bolded the day# and month# at the vaccination stage. Please see Supplementary Figure 2.

Reviewer #2:

The authors describe the amplitude and durability of neutralising antibody titers obtained after a third dose of CoronaVac either 2 months or 8 months after the second dose. Due to the ongoing SARS-CoV-2 pandemic which may be transitioning to endemic status, understanding boostability and durability of responses with CoronaVac is of interest to the SARS-CoV-2 vaccine field.

As commented on by the authors in the discussion, it is somewhat surprising that adults aged 60+ had higher titers after boosting than younger adults; likewise the decline is less than might be expected.

We would like to thank the reviewer for reviewing our manuscript; we are glad that she/he believes that our work could improve understanding of boostability and durability of responses with CoronaVac.

1. The primary limitation of the study is that while titers to the ancestral lineage of SARS-CoV-2 are presented, titers against variants of concern are not. As there is mounting evidence that titers are lower against VOCs with ancestral strain-based vaccines, this is an important element for understanding the data in context.

Thanks very much for pointing out this issue. We agree with the reviewer that immunogenicity assessment for variants is an important element for understanding the data in context under the current scenario in which variants of concern (VOCs) are spreading rapidly across the globe, and Omicron variants have become the predominately circulating strains. However, due to repeated freezing and thawing, and insufficient sera, we were unable to perform neutralization tests on specimens against a currently circulating virus. Previous study (*Khoury D et al, Nat Med, 2021*) reveals that high neutralizing titres against wild type are believed to be important for protection against novel circulating SARS-CoV-2 variants that potentially can lead to immune escape. Accordingly, our results of immunogenicity of booster doses against

wild type could indirectly reveal the relevant immunogenicity against VOCs. We have cited research on neutralization tests against variants of concern in vitro and addressed this limitation in the Discussion section.

2. Line 64 statement needs to be supported with a reference or softened to "a commonly used vaccine."

Thanks for pointing out this issue. We have revised the statement accordingly, please see as follows (lines 67-71):

“As CoronaVac is a commonly used vaccine and is contributing to the fight against the pandemic, assessing the duration of immunity following booster dose administration will be important for improving and updating immunization strategies.”

3. Line 96 clarity - replace dash with 3 to 4

Revised accordingly.

4. Line 94-95 correct carriage return

Revised accordingly.

5. Line 110 replace "durable" with "still elevated," likewise on line 167 replace "durable" with "maintained"

Revised accordingly.

6. Line 254 - typo "continuously" not "continuous".

Revised accordingly.

7. Lines 254-257 are political opinion. Please remove.

Removed accordingly.

Reviewers' Comments:

Reviewer #1:

Remarks to the Author:

Major point 1: The three tables provided additional info, but would serve as a preclinical assay per FDA guideline. That said, it will be nice to incorporate the three tables into supplemental material, so that the scientific community would know the data behind the status of the assay.

Major point 3: Have the virus stocks been deep sequenced to exclude any mutations, especially at the furin cleavage site? If so, please specify and add the info to the supplemental method session.

Major point 7a: "Samples with complete CPE in higher dilutions with no CPE in lower dilutions were assigned to an intermediate value between these two dilutions, which is why there were some titre values between 4 and 8." Wouldn't this practice inflate the value of nAb titer? A detailed data analysis session needs to be provided in the manuscript.

Major point 7c: "Since the standard operating procedures for neutralization test against SARS-CoV-2 has not been established, we defined the four-fold of lowest assigned neutralization titer (2) as the positive threshold according to experience with influenza vaccine. " It is unscientific to use experience from another virus and estimate to make a defined threshold for an important parameter (seropositive). This definition needs to be provided in the manuscript so that the scientific community can assess and understand the logic with the estimated seropositive cutoff based on estimate and experience.

There are diagnostic assays to find seronegative and seropositive samples in China, and actual experimental data are needed to define this threshold. This manuscript is a number game. However, I find the numbers/threshold are not solid sound.

Minor point 1: "Thanks very much for pointing out a mistake in the text. We revised the descriptions as "8 month after the 2nd dose" through the paper." The title of this ms is "six month follow up...", which is against this response. Please clarify and make corrections.

Reviewer #1:

Major point 1: The three tables provided additional info, but would serve as a preclinical assay per FDA guideline. That said, it will be nice to incorporate the three tables into supplemental material, so that the scientific community would know the data behind the status of the assay.

Response: Thanks for your comment. As suggested, we have incorporated the three tables into supplemental method session to provide additional information.

Major point 3: Have the virus stocks been deep sequenced to exclude any mutations, especially at the furin cleavage site? If so, please specify and add the info to the supplemental method session.

Response: Thanks for your comment. The virus used in the neutralization was passaged from the original vaccine strain.¹ Second generation sequencing for the Spike protein coding was conducted and the results showed complete agreement between the two generations, thus excluding the presence of mutations that may affect neutralizing potency. We think that readers will also be interested in this evaluation so we have added the information in the supplemental's methods section.

In addition to the deep sequencing analysis, to ensure the quality of the virus stock, neutralization tests of at least one batch of virus stock were conducted using positive control sera, and the results were compared with results of reference vaccine antigen testing. To be qualified, the test results of the ED50 and GMT must be within the verified qualified range compared with similar released virus seeds. We found that the virus was qualified. Finally, the working seed stock is titrated once every two years, testing three times, and using the mean value obtained by two staff testing simultaneously. If the virus titer is found to be reduced by 50% or more, it will not be used.

Major point 7a: "Samples with complete CPE in higher dilutions with no CPE in lower dilutions were assigned to an intermediate value between these two dilutions, which is why there were some titre values between 4 and 8." Wouldn't this practice inflate the value of nAb titer? A detailed data analysis session needs to be provided in the manuscript.

Response: Thanks for your comment. We conducted two sensitivity analyses to determine whether our assignment of an intermediate value between two dilutions will inflate or deflate the values of nAb titer. The first was to use only the higher dilution and the second was to use only the lower dilution when there were 2 adjacent wells with different dilutions. Sensitivity analysis results for the study cohorts are shown in the four tables below, in which “main analysis” refers to the intermediate values and “sensitivity analysis” refers to lower (tables 1 and 3) or higher (tables 2 and 4) values.

Table 1. Sensitivity analysis using lower dilution in adults aged 18-59 years old in cohort 2b-28d-8m

Visit	3 µg group			6 µg group		
	Main analysis	Sensitivity analysis	p-value	Main analysis	Sensitivity analysis	p-value
Baseline	2.0 (2.0-2.0)	2.0 (2.0-2.0)	-	2.0 (2.0-2.0)	2.0 (2.0-2.0)	-
Day 28 after dose 2	49.1 (40.1-60.2)	42.1 (34.3-51.7)	0.29	73.6 (60.2-90.0)	58.9 (48.1-72.1)	0.12
Day 180 after dose 2	6.7 (5.2-8.6)	5.8 (4.5-7.4)	0.42	7.0 (5.5-8.8)	6.0 (4.8-7.5)	0.34
Day 28 after dose 3	143.3 (112.3-182.8)	119.7 (93.7-153.1)	0.30	230.9 (171.2-311.5)	177.2 (131.4-239.0)	0.21
Day 180 after dose 3	36.4 (28.7-46.1)	30.3 (23.8-38.6)	0.28	62.8 (47.1-83.8)	54.9 (41.2-73.1)	0.50

Table 2. Sensitivity analysis using higher dilution in adults aged 18-59 years old in cohort 2b-28d-8m

Visit	3 µg group			6 µg group		
	Main analysis	Sensitivity analysis	p-value	Main analysis	Sensitivity analysis	p-value
Baseline	2.0 (2.0-2.0)	2.0 (2.0-2.0)	-	2.0 (2.0-2.0)	2.0 (2.0-2.0)	-
Day 28 after dose 2	49.1 (40.1-60.2)	54.8 (44.4-67.6)	0.46	73.6 (60.2-90.0)	86.3 (70.1-106.2)	0.27
Day 180 after dose 2	6.7 (5.2-8.6)	7.4 (5.7-9.7)	0.58	7.0 (5.5-8.8)	7.8 (6.1-10.0)	0.52
Day 28 after dose 3	143.3 (112.3-182.8)	162.7 (126.7-209.0)	0.47	230.9 (171.2-311.5)	278.7 (205.2-378.4)	0.38
Day 180 after dose 3	36.4 (28.7-46.1)	41.4 (32.6-52.7)	0.44	62.8 (47.1-83.8)	69.1 (51.5-92.9)	0.64

Table 3. Sensitivity analysis using lower dilution in adults aged 60 years and older in cohort 3-28d-8m

Visit	1.5 µg group			3 µg group			6 µg group		
	Main analysis	Sensitivity analysis	p-value	Main analysis	Sensitivity analysis	p-value	Main analysis	Sensitivity analysis	p-value
Baseline	2.0 (2.0-2.0)	2.0 (2.0-2.0)	-	2.0 (2.0-2.1)	2.0 (2.0-2.0)	-	2.1 (1.9-2.2)	2.1 (1.9-2.2)	
Day 28 after dose 2	23.4 (19.4-28.3)	19.0 (15.7-23.0)	0.12	41.2 (34.2-49.6)	32.2 (26.8-38.8)	0.06	49.9 (42.2-58.9)	38.7 (32.7-45.9)	0.04
Day 180 after dose 2	3.1 (2.7-3.6)	2.9 (2.6-3.3)	0.55	3.4 (2.9-4.1)	3.3 (2.8-3.9)	0.74	4.1 (3.3-5.0)	3.7 (3.0-4.5)	0.51
Day 28 after dose 3	99.6 (60.6-163.5)	86.1 (52.0-142.6)	0.68	158.5 (96.9-259.2)	125.0 (76.1-205.2)	0.49	178.9 (123.2-259.9)	137.9 (96.1-197.8)	0.31
Day 180 after dose 3	20.6 (15.5-27.3)	17.4 (13.2-23.0)	0.40	53.2 (39.7-71.1)	42.4 (31.8-56.4)	0.27	91.2 (71.5-116.3)	74.1 (57.9-94.7)	0.23

Table 4. Sensitivity analysis using higher dilution in adults aged 60 years and older in cohort 3-28d-8m

Visit	1.5 µg group			3 µg group			6 µg group		
	Main analysis	Sensitivity analysis	p-value	Main analysis	Sensitivity analysis	p-value	Main analysis	Sensitivity analysis	p-value
Baseline	2.0 (2.0-2.0)	2.0 (2.0-2.0)	-	2.0 (2.0-2.1)	2.0 (2.0-2.1)	-	2.1 (1.9-2.2)	2.1 (1.9-2.2)	-
Day 28 after dose 2	23.4 (19.4-28.3)	27.2 (22.4-32.9)	0.28	41.2 (34.2-49.6)	49.0 (40.5-59.4)	0.20	49.9 (42.2-58.9)	59.6 (50.4-70.6)	0.13
Day 180 after dose 2	3.1 (2.7-3.6)	3.2 (2.7-3.8)	0.70	3.4 (2.9-4.1)	3.5 (2.9-4.3)	0.82	4.1 (3.3-5.0)	4.3 (3.5-5.5)	0.67
Day 28 after dose 3	99.6 (60.6-163.5)	110.3 (67.1-181.6)	0.77	158.5 (96.9-259.2)	187.6 (114.2-308.2)	0.62	178.9 (123.2-259.9)	215.3 (145.5-318.4)	0.49
Day 180 after dose 3	20.6 (15.5-27.3)	23.2 (17.4-31.0)	0.56	53.2 (39.7-71.1)	62.4 (46.4-84.1)	0.44	91.2 (71.5-116.3)	105.7 (82.6-135.2)	0.40

When we made comparisons between the main analysis values and the sensitivity analysis values, we found no significant differences. Based on the sensitivity analyses, we continue to use the intermediate value in the main analysis. We have added these sensitivity analyses to the supplementary material to assure the readers of the appropriateness of our use of intermediate values.

Major point 7c: "Since the standard operating procedures for neutralization test against SARS-CoV-2 has not been established, we defined the four-fold of lowest assigned neutralization titer (2) as the positive threshold according to experience with influenza vaccine. " It is unscientific to use experience from another virus and estimate to make a defined threshold for an important parameter (seropositive). This definition needs to be provided in the manuscript so that the scientific community can assess and understand the logic with the estimated seropositive cutoff based on estimate and experience.

There are diagnostic assays to find seronegative and seropositive samples in China, and actual experimental data are needed to define this threshold. This manuscript is a number game. However, I find the numbers/threshold are not solid sound.

Response: Thanks very much for pointing out this issue. We agree that defining the threshold based on experience from other virus is a limitation of our study. To determine a definition of seropositivity for COVID-19 vaccine studies, we reviewed publications and preprints that reported clinical trial results of COVID-19 vaccines that were identified by the search term, "(COVID-19 OR SARS-CoV-2) AND vaccin* AND (immunogenic* OR "immune response") AND trial". The seropositive thresholds used in live virus neutralization assays from the retrieved literature are listed in the table below.

Ref	Neutralization test	Detection limit	Seropositive threshold	DOI
Pu et al	Microtitration assay	Not provided	1:4	10.1016/j.vaccine.2021.04.006
Sadoff et al	Microtitration assay	58 IC50	58 IC50	10.1101/2021.08.25.21262569
Ishmukhametov et al	Virus neutralization test	1:8	1:8	10.1101/2022.02.08.22270658
Che et al	Microtitration assay	Not provided	1:4	10.1093/cid/ciaa1703
Melo-Gonzalez et al	Conventional plaque-reduction neutralization test	1:4	1:4	10.3389/fimmu.2021.747830
Pan et al	Micro cytopathogenic effect assay	1:4	1:4	10.1097/CM9.0000000000001573
Liu et al	Microtitration assay	1:4	1:4	10.1093/infdis/jiab627
Chen et al	Microneutralization assay	1:4	1:4	10.1016/S2666-5247(21)00280-9
Fadlyana et al	Neutralization of antibody assay	1:4	1:4	10.1016/j.vaccine.2021.09.052
Sadoff et al	Microneutralization assay	58 IC50	58 IC50	10.1056/NEJMoa2034201
Feng et al*	Micro cytopathogenic effect assay	1:4	1:32, 64, 128, 256*	10.1186/s40249-021-00924-2

* Feng et al used four different seropositive cut-off values to calculate positive rates

Most of the reports listed above used the detection limit as seropositive threshold. However, we believe that using the lower limit of detection as the seropositivity cutoff is unreasonable because seropositivity should be correlated with potential protection and should be defined by diagnostic assays and detailed experimental data.

We also agree with the reviewer's point that using the experience from other viruses to define this positive threshold could be considered unscientific. In view of our main study objective, which was to evaluate antibody levels six months after three doses of CoronaVac vaccine, and to make our reporting both consistent with previous publications and accurate, we have removed the statements and results associated with seropositive threshold and seropositivity rates from the manuscript and retained the protective threshold (1:33) to indicate the potential protection effect.²

Minor point 1: "Thanks very much for pointing out a mistake in the text. We revised the descriptions as "8 month after the 2nd dose" through the paper." The title of this ms is "six month follow up...", which is against this response. Please clarify and make corrections.

Response: Thanks for your comment. We should clarify that 8 months indicated the interval between the booster dose and the 2nd dose (completion of two-dose primary series) and "six-month follow-up" in the title indicates the follow-up period after booster doses. The entire follow-up period, from the 1st dose (day 0) was 15 months for cohort 2b-28d-8m and cohort 3-28d-8m.

References

1. Gao, Q., *et al.* Development of an inactivated vaccine candidate for SARS-CoV-2. *Science* **369**, 77-81 (2020).
2. Khoury, D.S., *et al.* Neutralizing antibody levels are highly predictive of immune protection from symptomatic SARS-CoV-2 infection. *Nature medicine* (2021).